# Comparative Behavioral Phenotypes of *Fmr1* KO, *Fxr2* Het, and *Fmr1* KO/*Fxr2* Het Mice

**DOI:** 10.3390/brainsci9010013

**Published:** 2019-01-16

**Authors:** Rachel Michelle Saré, Christopher Figueroa, Abigail Lemons, Inna Loutaev, Carolyn Beebe Smith

**Affiliations:** Section on Neuroadaptation and Protein Metabolism, National Institute of Mental Health, National Institutes of Health, Department of Health and Human Services, Bethesda, MD 20814, USA; Rachel.Sare@nih.gov (R.M.S.); Christopher.Figueroa@nih.gov (C.F.); Abigail.Lemons@nih.gov (A.L.); Loutaev.inna@nih.gov (I.L.)

**Keywords:** Fragile X, FMRP, *Fxr2*, *Fmr1*

## Abstract

Fragile X syndrome (FXS) is caused by silencing of the *FMR1* gene leading to loss of the protein product fragile X mental retardation protein (FMRP). FXS is the most common monogenic cause of intellectual disability. There are two known mammalian paralogs of FMRP, FXR1P, and FXR2P. The functions of FXR1P and FXR2P and their possible roles in producing or modulating the phenotype observed in FXS are yet to be identified. Previous studies have revealed that mice lacking *Fxr2* display similar behavioral abnormalities as *Fmr1* knockout (KO) mice. In this study, we expand upon the behavioral phenotypes of *Fmr1* KO and *Fxr2*^+/−^ (Het) mice and compare them with *Fmr1* KO/*Fxr2* Het mice. We find that *Fmr1* KO and *Fmr1* KO/*Fxr2* Het mice are similarly hyperactive compared to WT and *Fxr2* Het mice. *Fmr1* KO/*Fxr2* Het mice have more severe learning and memory impairments than *Fmr1* KO mice. *Fmr1* KO mice display significantly impaired social behaviors compared to WT mice, which are paradoxically reversed in *Fmr1* KO/*Fxr2* Het mice. These results highlight the important functional consequences of loss or reduction of FMRP and FXR2P.

## 1. Introduction

Fragile X syndrome (FXS) is the leading heritable cause of intellectual disability in humans, affecting about 1 in 4000 males [1]. In addition to intellectual disability, FXS patients often display multiple behavioral phenotypes including hyperactivity, attention deficits, susceptibility to seizures, hypersensitivity, sleep abnormalities, and social anxiety/autism-like behaviors [2]. FXS is primarily caused by a CGG repeat expansion in the 5′UTR of *FMR1* which leads to gene silencing and the consequent loss of its protein product, FMRP [3]. FMRP is an RNA-binding protein with over 800 mRNA targets [4]. FMRP is highly expressed in the brain, and is thought to act as a translational suppressor [5]. The loss of translational regulation by FMRP has been shown to lead to excessive brain protein synthesis in *Fmr1* knockout (KO) mice, a mouse model of FXS [6]. In addition to its presumed role in mRNA regulation, FMRP is also thought to be involved in nuclear export and cytoplasmic transport [7], ion channel activity [8], and participates in the DNA damage response [9].

*Fmr1* KO mice exhibit many of the behavioral symptoms seen in humans with FXS, including hyperactivity, deficits in learning and memory, reduced preference for social novelty, repetitive behaviors, and reduced sleep [10,11,12]. In mammals, there are two autosomal paralogs of FMRP, FXR1P and FXR2P, which together comprise the fragile-X related (FXR) family of proteins [13]. FXR1P and FXR2P share a conserved structure and amino acid sequence (86% and 70% respective conservation in the N-terminus and central regions) with FMRP; this conservation includes the presumed RNA-binding sites [13]. Due to sequence homology and similar expression patterns, these proteins are hypothesized to have similar functions [14]. Therefore, it is thought that these proteins can, at least partially, compensate for loss of FMRP, thereby playing an as yet undiscovered role in FXS [15]. The FXR family of genes has been investigated in a clinical setting. The *FXR2* gene is located on chromosome 17. Microdeletions in the 17p13.1 comprise a syndrome that is often associated with dysmorphic features and developmental delay [16,17]. Given that this microdeletion syndrome is accompanied by the loss of multiple genes, it is hard to identify the specific role of *FXR2*. Additionally, it has been suggested that accumulation of single nucleotide polymorphisms in the fragile x gene family (including *FMR1*, *FXR1*, and *FXR2*) are associated with autistic phenotypes [18].

To further study the roles of the FXR mutations, *Fxr1* and *Fxr2* KO mice have been generated [15,19]. *Fxr1* KO is lethal shortly after birth [19]. In addition to brain expression, FXR1P is highly expressed in cardiac and skeletal muscle (more so than FMRP or FXR2P) and is likely involved in RNA translation/transport regulation in muscle; it is this role that may lead to early lethality in knockout models [19]. Like FMRP, FXR2P is expressed in brain with a similar regional distribution and mice deficient in FXR2P (*Fxr2* KO) exhibit some of the same behavioral deficits seen in *Fmr1* KO mice such as hyperactivity and learning and memory impairments [15].

*Fmr1*/*Fxr2* double KOs animals have been shown to have a greater enhancement in metabotropic glutamate receptor activated long-term depression than *Fmr1* or *Fxr2* KO mice [20]. Behaviorally, *Fmr1*/*Fxr2* double KOs exhibited increased hyperactivity and greater impairments in contextual fear conditioning compared to single *Fmr1* or *Fxr2* KO mice [21]. Importantly, it was noted that *Fmr1*/*Fxr2* double KO mice had reduced survival rates and were often runted in size [21], so the adult animals studied were a select population of survivors with likely higher functioning than the total population. To get a better idea of the overlapping and novel functions of FMRP and FXR2P, we chose to perform studies in *Fmr1* KO/ *Fxr2* Het mice to ensure a non-biased sample. Both *Fmr1* KO/*Fxr2* KO and *Fmr1* KO/*Fxr2* Het have been shown to have similar impairments in circadian rhythm compared to *Fmr1* KO mice [22], indicating that, in this modality, *Fxr2* haploinsufficiency is sufficient to induce an impairment in the context of *Fmr1* deletion. Furthermore, *Fmr1* KO/*Fxr2* Het mice had a further decrease in sleep than *Fmr1* KO animals [11].

In this manuscript, we present behavioral studies of *Fmr1* KO, *Fxr2* Het, and *Fmr1* KO/*Fxr2* Het mice. We hypothesized that *Fmr1* KO/*Fxr2* Het mice would have an exaggerated behavioral phenotype compared to either single mutation. We found that both *Fmr1* KO and *Fmr1* KO/*Fxr2* Het mice are similarly hyperactive compared to wild-type (WT) and *Fxr2* Het mice. Additionally, *Fmr1* KO mice have statistically significantly impaired social behavior compared with WT mice, which is paradoxically reversed in *Fmr1* KO/*Fxr2* Het mice.

## 2. Materials and Methods

### 2.1. Animals

These studies were conducted on male WT, *Fmr1* KO, *Fxr2* Het, and *Fmr1* KO/*Fxr2* Het mice on a C57BL/6J background maintained in house. The original breeders were obtained from David Nelson at Baylor [22] and genotyped as previously described [11]. Breeders were either male *Fxr2* Het and female *Fmr1* Het mice or male WT mice and female *Fmr1* Het/*Fxr2* Het mice. Mice were group housed in a central climate-controlled facility with a standard 12:12 light:dark (lights on at 6:00 a.m.) facility. Food and water were available ad libitum. All procedures were carried out in accordance with the National Institutes of Health Guidelines on the Care and Use of Animals and approved by the National Institute of Mental Health Animal Care and Use Committee.

### 2.2. Behavior Testing

Mice were subjected to a battery of behavior tests starting between 63–77 days of age. Behavior tests were conducted from the least stressful to the most stressful in the following order: open field, novel object recognition, zero maze, marble burying, social behavior, and passive avoidance. No more than two tests were conducted in the same week with at least two days between tests. Our initial cohort of animals consisted of WT (*n* = 33), *Fmr1* KO (*n* = 26), *Fxr2* Het (*n* = 22), and *Fmr1* KO/ *Fxr2* Het (*n* = 24).

### 2.3. Open Field

Open field testing was done to assay activity and anxiety-like behaviors in response to a novel environment. Mice were placed in the center of a plexiglass open field arena (Coulbourn Instruments, Holliston, MA, USA) and allowed to explore for 30 min. Total distance traveled as well as distance traveled in the center of the chamber were recorded by TruScan software (Coulbourn Instruments) in five-minute epochs. Anxiety-like behavior was assessed by the ratio of distance traveled in the center to the total distance traveled; the center to total distance ratio is inversely proportional to anxiety levels. For open field testing, nine WT mice were not included (one due to missing data, three due to equipment malfunction, and five were statistical outliers); eight *Fmr1* KO mice were not included (one due to missing data, three due to equipment malfunction, and four were statistical outliers); eight *Fxr2* Het mice were not included (three due to equipment malfunction and six were statistical outliers); and ten *Fmr1* KO/ *Fxr2* Het mice were not included (two due to missing data, three due to equipment malfunction, and five were statistical outliers.

### 2.4. Novel Object Recognition (NOR)

NOR was performed to assay learning and memory capability. Testing was performed in the open field arena (Coulbourn Instruments). We used open field testing as the habituation phase for the NOR on Day 1. On Days 2 and 3, two identical objects were placed in the arena and the mouse was allowed to explore the objects for 5 min. Any animal that showed a preference for one of the two identical objects on these training days was eliminated from the study. Any animal that did not spend more than 10 seconds sniffing objects was also eliminated. On Day 4, one of the objects was replaced by a novel object and the mouse was allowed to explore for 5 min. The behavior of the mice was recorded by a video camera for later analysis of sniffing behavior. A discrimination index was calculated as the (sniffing time of novel—sniffing time of familiar)/total sniffing time. For NOR, 14 WT mice were not included (one was not run due to scheduling conflicts and 13 did not sniff for enough time); seven *Fmr1* KO mice were not included (one was not run due to scheduling conflicts, four did not sniff for enough time, and two knocked over the objects); seven *Fxr2* Het mice were not included (six did not sniff for enough time and one knocked over the objects); and seven *Fmr1* KO/*Fxr2* Het mice were not included (five were not run because of scheduling conflicts, one demonstrated a side preference, and one did not sniff for enough time).

### 2.5. Zero Maze

Anxiety-like behavior was assayed by means of the zero maze. Mice were placed inside the closed portion of the maze facing toward the open portion. They were then allowed to explore the maze for 5 min and the total time spent in the open portion was recorded. A mouse was counted as being in the open/closed portion when both front paws had crossed the threshold. If the mouse fell off the maze during the test period, its results were disqualified from the analysis. For zero maze testing, five WT mice were not included (one due to missing data, two fell off, and two were statistical outliers); two *Fmr1* KO mice were not included (one due to missing data and one was a statistical outlier); 0 *Fxr2* Het mice were not included; and two *Fmr1* KO/*Fxr2* Het mice were not included due to missing data.

### 2.6. Marble Burying

Repetitive behaviors were assayed by means of the marble burying test. A standard cage with bedding at a depth of 4.5 cm was prepared and 20 marbles were evenly arranged in a 4 × 5 grid on top of the bedding. The test mouse was placed in the cage and allowed to explore for 30 min. The number of marbles buried (>50% covered) was recorded. For marble burying, four WT mice were not included (one due to missing data, one not properly set up, and two were outliers); three *Fmr1* KO mice were not included (one due to missing data, one was not tested due to scheduling issues, and one was a statistical outlier); one *Fxr2* Het was not included as it was a statistical outlier; and four *Fmr1* KO/*Fxr2* Het mice were not included (two due to missing data, one was not tested due to scheduling issues, and one was a statistical outlier).

### 2.7. Social Behavior

Sociability and preference for social novelty were tested by means of the three chamber social behavior apparatus [23]. The test mouse was placed in the center chamber of a three-chamber apparatus and allowed to explore all three chambers for a 5 min habituation period. If an animal spent more than 3 min in one chamber or did not explore a chamber during the habituation period, it was excluded. Following habituation, an empty holding enclosure (10 cm diameter) was introduced to one of the side chambers and another identical enclosure with a stranger mouse inside was introduced to the other side. The test mouse was allowed to explore for another 5 min. In the third part of the test, a novel stranger mouse was introduced to the previously empty enclosure. The test mouse was then allowed to explore for another 5 min. Time in each chamber was recorded automatically by photobeam breaks. Video recording was performed for later assessment of sniffing behavior which was assessed by means of an automated software (TopScan) (Clever Systems, Reston, VA, USA). Sniffing was defined as the animal being within 2.0 cm of the enclosure with his nose directed toward the enclosure. For social behavior, 12 WT mice were not included (one due to missing data, three because of a side preference during habituation, one was not run due to scheduling issues, and six were statistical outliers); seven *Fmr1* KO mice were not included (two due to missing data, one due to equipment malfunction, and three were statistical outliers); eight *Fxr2* Het mice were not included (one because of missing data, three because of a side preference during habituation and four were statistical outliers); eight *Fmr1* KO/*Fxr2* Het mice were not included (two due to missing data, three because of a side preference during habituation, and three were statistical outliers).

### 2.8. Passive Avoidance

The passive avoidance test was done to assess memory. The apparatus is a light/dark shuttle box with a shocker in the floor and a guillotine door (Coulbourn Instruments). The test was conducted over three consecutive days. On Day 1, habituation to the shuttle box, the test mouse was placed on the lighted side; after 30 s, the guillotine door opened to the dark chamber and the mouse was free to enter. Once the mouse entered the dark chamber, it was removed from the apparatus. Day 2 was the training day. The mouse was placed in the lighted chamber, and after 30 s the door opened to the dark chamber. Once the mouse entered the dark chamber, it received a mild foot shock (0.3 mA, 1 s). The mouse was then placed in a recovery cage for 2 min before being placed back in the lighted chamber with the guillotine door closed and the training repeated. On Day 3, the mouse was placed in the lighted chamber, and after 30 s the door opened to the dark chamber. The latency (maximum of 10 min) for the mouse to enter the dark chamber was recorded. If the animal did not enter the dark, the maximum value was assigned. For passive avoidance, 10 WT mice were not included (three were not run due to scheduling issues and seven due to equipment issues); six *Fmr1* KO were not included (three were not run due to scheduling issues and three due to equipment issues); six *Fxr2* Het were not included (three were not run due to scheduling issues and three due to equipment issues); seven *Fmr1* KO/*Fxr2* Het were not included (two were not run due to scheduling issues and five due to equipment issues).

### 2.9. Protein Expression

Adult male *Fmr1* KO and *Fmr1* KO/*Fxr2* Het mice were decapitated and brains were rapidly removed and the hippocampus dissected. Tissue was placed into Precellys lysis kits (Bertin Corportation, Rockville, MD, USA) and stored at −80 °C until further processing. Protein was extracted using the Precellys Homogenizer as previously described [24] and 15 µg of protein was loaded per lane. We used the Bio-Rad mini-protein stain-free gel technology for Western blotting as previously described [24]. We chose the stain free technology in order to normalize to total protein loaded rather than a standard housekeeping gene. In disorders like fragile X syndrome, where protein translation is thought to be affected, normalizing to a housekeeping protein might alter the results. The blot was incubated in primary antibody solution (1:1000 FXR2 (A303-894A) (Bethyl Laboratories, Montgomery, TX, USA)) overnight at 4 °C.

### 2.10. Statistical Analysis

Data reported are the means ± standard error of the mean (SEM). Data for a given behavioral test were excluded if one data point from that mouse was more than two standard deviations away from the mean. Zero maze, marble burying, and passive avoidance were analyzed by a one-way ANOVA with genotype as the variable. Open field (epoch) and social behavior (chamber) were analyzed by repeated measures ANOVA with the corresponding repeated measure as a within subjects’ variable. When appropriate, post-hoc *t*-tests were performed with Bonferroni correction. The results of all ANOVA tests are presented in Table 1. We considered tests with *p* ≤ 0.05 statistically significant. In the figures, these effects are denoted with a “*”. We also indicate effects that are approaching statistical significance 0.05 ≤ *p* ≤ 0.10 with a “~”.

## 3. Results

### 3.1. Expression of FXR2

We measured FXR2 protein expression in *Fmr1* KO and *Fmr1* KO/*Fxr2* Het mice. FXR2 protein expression was reduced by 60% in Fxr2 heterozygous animals compared to controls (*p* = 0.0001, student’s *t*-test) (Figure 1). Protein expression was normalized to total protein loaded.

### 3.2. Activity in the Open Field

We measured activity in response to a novel environment by analyzing distance traveled in the open-field test. There was no genotype × epoch interaction indicating that, regardless of genotype, all mice showed a typical burst of activity in the beginning of exposure to the open field followed by adaptation to the environment (measured by decreased activity with time) (Figure 2). We found a statistically significant main effect of genotype (*p* < 0.001) (Table 1). Both *Fmr1* KO (*p* = 0.017) and *Fmr1* KO/*Fxr2* Het mice (*p* < 0.001) were statistically significantly hyperactive compared to WT mice (Figure 2). In addition, *Fmr1* KO/*Fxr2* Het mice were more active compared to *Fxr2* Hets (*p* = 0.025). WT and *Fxr2* Het mice showed similar levels of activity (Figure 2).

### 3.3. Anxiety-Like Behavior

The ratio of distance traveled in the center to total distance traveled was analyzed as an inverse measure of anxiety-like behavior. We found a statistically significant (*p* = 0.005) genotype × epoch interaction (Table 1). Post-hoc tests are shown in Table 2. In general, both *Fmr1* KO and *Fmr1* KO/*Fxr2* Het mice traveled more relative distance in the center compared to WT indicating lower anxiety. *Fxr2* Het mice had similar distance traveled in the center compared to WT mice (Figure 3A). These results suggest that both *Fmr1* KO and *Fmr1* KO/*Fxr2* Het mice are statistically significantly less anxious than either WT or *Fxr2* Hets.

We also used the time spent in the open portion of the zero maze as another inverse measure of anxiety-like behavior. We found that the main effect of genotype approached statistical significance (*p* = 0.084). Post-hoc t-tests showed a trend for the *Fmr1* KO/*Fxr2* Het mice to spend more time in the open portion than WT mice (*p* = 0.065) (Figure 3B). These results are consistent with a reduced anxiety phenotype also demonstrated in the open field in *Fmr1* KO/*Fxr2* Het mice.

### 3.4. Repetitive Behavior

The number of marbles buried during this 30 min test is considered a measure of repetitive behavior. We found a near statistically significant effect of genotype (*p* = 0.052) (Table 1). Post-hoc comparisons revealed trends with *Fmr1* KO mice burying more marbles than both WT (*p* = 0.095) and *Fmr1* KO/*Fxr2* Hets (*p* = 0.098) (Figure 4). These trends suggest that, of these four genotypes, only *Fmr1* KO mice show elevated repetitive behavior.

### 3.5. Social Behavior

We tested social behavior by means of the three-chambered apparatus. In this test, we measured both time in chamber and time sniffing either the enclosure or the mouse. In the first phase of the task, the sociability phase, all genotypes showed a preference for the stranger mouse compared to the object with respect to both the time in chamber (Figure 5A) and sniffing time (Figure 5B) (Table 1).

For the second phase of the task, the preference for social novelty, all four genotypes spent about the same amount of time in the two chambers showing no preference for either the novel or familiar mouse (Figure 5C) (Table 1). The time sniffing either the familiar or novel mouse differentiated the genotypes (Figure 5D). The genotype × chamber interaction was statistically significant (*p* = 0.043) (Table 1), and post-hoc t-tests showed that *Fmr1* KO/*Fxr2* Het (*p* = 0.006) mice showed a clear preference for the novel mouse compared with the familiar mouse. This phenotype was also seen in WT, but to a lesser degree (*p* = 0.062). Sniffing times for the novel and familiar mice in *Fmr1* KO and *Fxr2* Het mice were not significantly different, suggesting no preference for social novelty in these genotypes (Figure 5D).

### 3.6. Learning and Memory

To assay learning and memory, we performed NOR. Our data had high variability and we did not see any statistically significant effects (Table 1, Figure 6).

For learning and memory, we also performed passive avoidance testing. With this test, we found a statistically significant main effect of genotype (*p* = 0.022) (Table 1). Post-hoc *t*-tests indicate that *Fmr1* KO/*Fxr2* Het mice had statistically significantly (*p* = 0.014) shorter latencies to enter the dark than WT mice. This result suggests that learning and memory is more compromised in mice with the double mutation (Figure 7).

## 4. Discussion

Here, we present behavioral similarities and differences between WT, *Fmr1* KO, *Fxr2* Het, and *Fmr1* KO/*Fxr2* Het mice. Our data suggest that, in the context of *Fmr1* deletion, deleting one copy of *Fxr2* does not have a detrimental effect on activity and therefore one copy may be sufficient for maintaining this behavior. With respect to learning and memory, deleting both *Fmr1* and *Fxr2* has a more severe effect than either single mutation suggesting that one protein can compensate for loss of the other. Finally, paradoxically, these proteins seem to have opposite roles in social behavior. Deleting both *Fmr1* and one copy of *Fxr2* results in improvements in behavior compared to single *Fmr1* deletion. These data highlight possible differing roles of these proteins depending on the behavior examined. Our results add to our understanding of the functions of fragile X related proteins.

*Fmr1* KO/*Fxr2* KO mice have been previously described to have worse learning and memory (measured by fear conditioning) than either of the single mutations alone [21]. In passive avoidance, we find that *Fmr1* KO/*Fxr2* Het mice have more profound learning and memory impairments than do the other genotypes studied. This behavior may be a reflection of either learning and memory or of impulsivity [25] and impulsivity is also known to be affected in *Fmr1* KO mice [26]. We did not find deficiencies in the passive avoidance test in single *Fmr1* KO mice, which contrasts with several published studies [12]. Importantly, whereas we do not see statistically significant genotype differences between WT and *Fmr1* KO mice, the mean values do reflect the expected trend that *Fmr1* KO mice have shorter latencies to enter the dark suggesting impaired memory. The differences between this and previous studies might have more to do with increased variability in the current results. During the course of this study, we had to replace our passive avoidance equipment. Although our new equipment was from the same company (Coulbourn Instruments), it is possible that changing the system affected the results. Another possible caveat is that we capped the latency to enter the dark compartment at 10 min. A large number of animals timed out of the study and were assigned the maximum value of 10 min (43.5% WT, 12.5% *Fxr2* Het, 35% *Fmr1* KO, 5.9% *Fmr1* KO/*Fxr2* Het). If the threshold were higher, it might be possible to detect more genotype differences.

One important note to the behavior testing conducted in this study is the fact that two different experimenters performed the testing, one male and one female. Prior studies have shown that the sex of the experimenter can have effects on anxiety behavior in WT mice. The mice respond with increased anxiety in the presence of a male experimenter [27], and effects on anxiety may have affected other behavioral measures. It is not known how *Fmr1* KO and *Fmr1* KO/*Fxr2* Het mice may respond to this difference, but we note that in our study, the male experimenter tested more *Fmr1* KO/*Fxr2* Het mice than the female experimenter. Whereas this may have biased our results, the fact that *Fmr1* KO/*Fxr2* Het mice demonstrated the least anxiety-related behavior suggests that the direction of the bias would be to underestimate the genotype difference.

Previously published results suggested that anxiety-like behavior was the same between the *Fmr1* KO and *Fmr1* KO/*Fxr2* KO mice based on the open field test and the light:dark box [21]. Similarly, our open field results suggest that *Fmr1* and *Fmr1* KO/*Fxr2* Het mice have similar reductions in anxiety compared to WT mice. However, another measure of anxiety-like behavior, the zero maze, showed a statistical trend indicating that *Fmr1* KO/*Fxr2* Het mice had even lower anxiety than *Fmr1* KO mice. It should be noted that, although studies have reported a phenotype in *Fmr1* KO mice on the zero maze [12], we did not detect any statistically significant differences between WT and *Fmr1* KO mice in the current study.

We also assessed repetitive behaviors and social behaviors, two behaviors that are relevant to mouse models of autism. This is important given that autism is reported to be present in about 50% of patients with fragile X syndrome [28]. In agreement with previous studies [12], we found that *Fmr1* KO mice had a trend toward increased repetitive behaviors and impaired preference for social novelty. Paradoxically, these phenotypes were reversed in *Fmr1* KO/*Fxr2* Het mice, suggesting contrasting functions between these two proteins in repetitive and social behaviors.

Although FMRP and FXR2P have similar regional distributions in brain, the results of our studies indicate that the functional roles of these proteins may differ. Other studies have also indicated functional differences between FMRP and FXR2P. For example, at the synapse, the fragile X related proteins form granules with ribonucleoprotein particles [29]. FXR2P is always expressed in these granules, but FMRP is only co-expressed in the granules in certain brain regions (the granules in much of the brain stem and cerebellum do not contain FMRP) [30]. For the most part, FMRP and FXR2P seem to have overlapping, cooperative roles in regulating metabolism. It had been shown previously that the phenotype of *Fmr1* KO/*Fxr2* Het mice was not as severe as that of *Fmr1* KO/*Fxr2* KO mice [31]. To fully assess the potentially compensatory functions of these proteins for each other, it would be best to study full *Fxr2* KOs with and without the deletion of *Fmr1*. However, we chose to study only the *Fxr2* Het because we know from prior studies that these animals have a high mortality and we were concerned that we would be studying a selected population of survivors.

## 5. Conclusions

Overall, our data point to important functions of *Fmr1* and *Fxr2* in behavior. Based on these results, there are some domains (like learning and memory) where *Fmr1* and *Fxr2* may have overlapping functions and can partially compensate for loss of the other. However, there are other domains (like social behavior) in which they appear to have different, even opposite roles in behavior.

## Figures and Tables

**Figure 1 brainsci-09-00013-f001:**
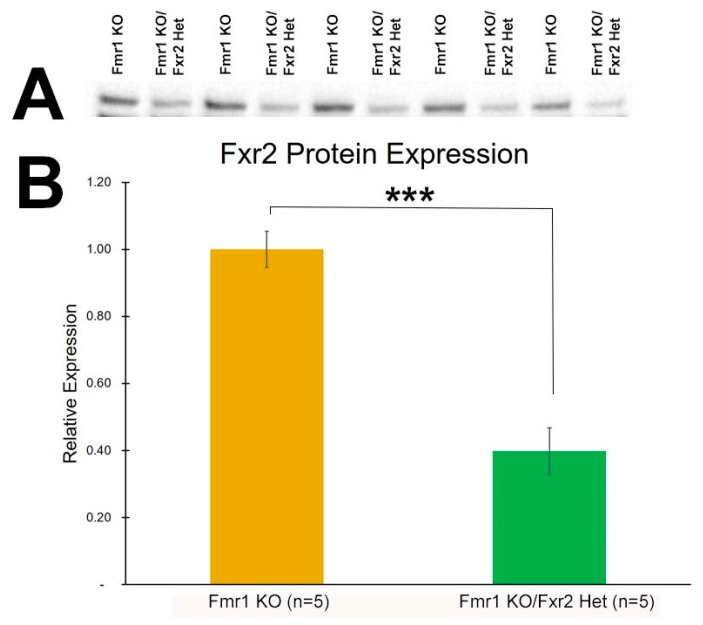
FXR2 protein expression is reduced in *Fxr2* heterozygous mice. (**A**) Western blots showing FXR2 protein expression in *Fmr1* KO and *Fmr1* KO/*Fxr2* Het hippocampus. (**B**) FXR2 protein expression is reduced by 60% in *Fmr1* KO/*Fxr2* Het animals (*** *p* = 0.0001, student’s *t*-test). Each bar represents the mean ± SEM for the number of mice indicated on the figure. (**C**) Stain-free image of total protein loaded for FXR2 protein expression. FXR2 protein expression (seen in Figure 1) was normalized to the total protein loaded shown here. The image was acquired by UV Trans illumination, exposed for 2 s.

**Figure 2 brainsci-09-00013-f002:**
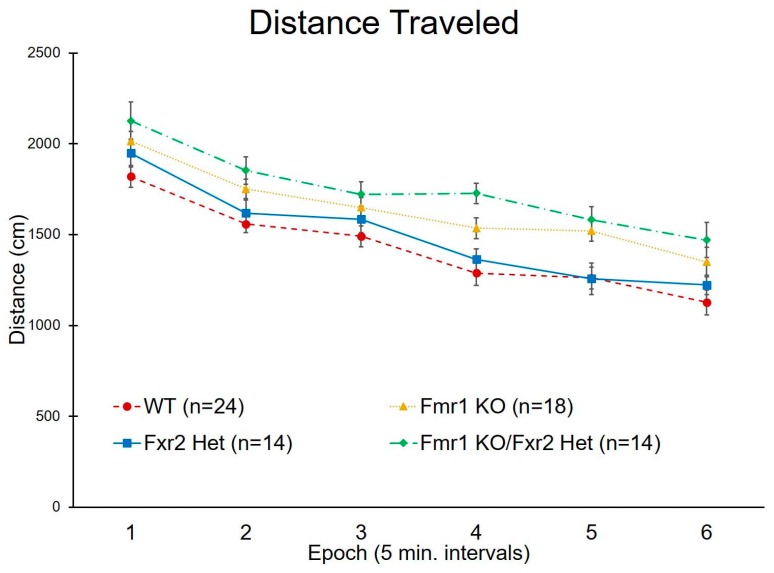
Distance traveled in the open field across the 30 min testing period. We found a statistically significant main effect of genotype (*p* < 0.001). Both *Fmr1* KO and *Fmr1* KO/*Fxr2* Het mice were statistically significantly hyperactive compared to WT (*p* = 0.017 and *p* < 0.001, respectively). Additionally, *Fmr1* KO/*Fxr2* Het mice were more active than *Fxr2* Het mice (*p* = 0.025). Each point represents the mean ± SEM for the number of mice indicated on the figure.

**Figure 3 brainsci-09-00013-f003:**
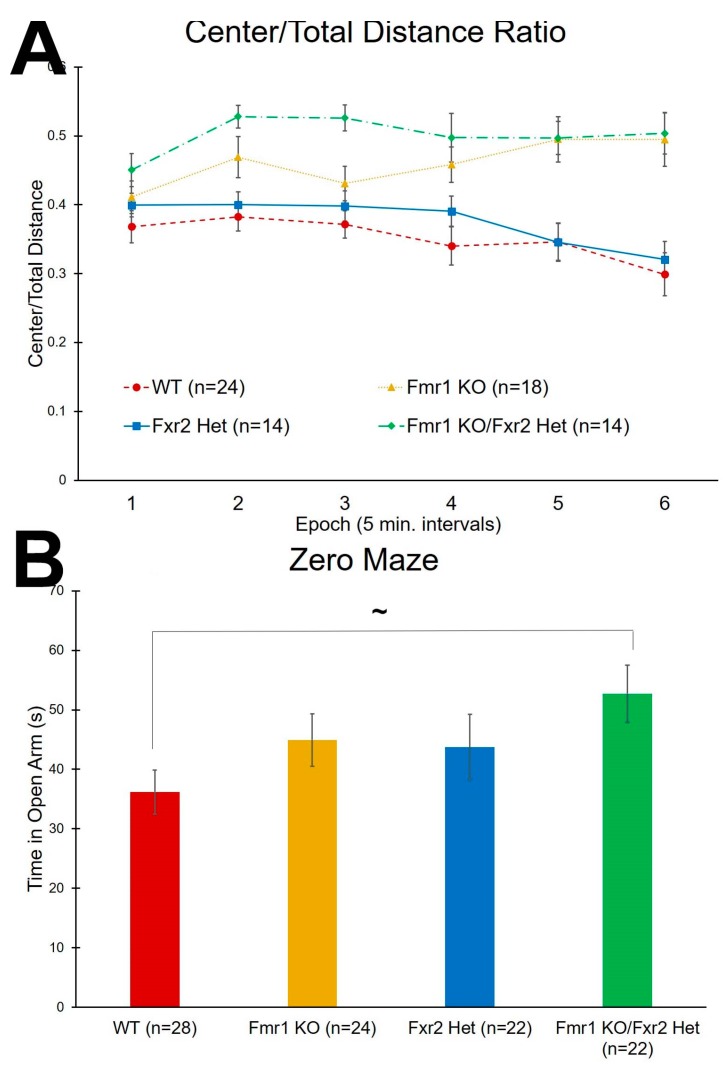
Tests for anxiety-like behavior. (**A**) In the open field test, the ratio of center distance to total distance traveled is an inverse measure of anxiety-like behavior. The genotype × epoch interaction was statistically significant (*p* = 0.005). Overall, *Fmr1* KO/*Fxr2* Het mice traveled more relative distance in the center than WT mice throughout the test. Similarly, *Fmr1* KO also traveled more relative distance in the center than WT mice in almost every epoch of the test (Table 2). *Fmr1* KO (Epochs 5 and 6) and *Fmr1* KO/*Fxr2* Het mice (Epochs 2, 3, 5, and 6) also traveled more relative distance in the center than *Fxr2* Het mice. In epoch 3, *Fmr1* KO/*Fxr2* Het mice traveled more relative distance in the center than *Fmr1* KO mice. Each point represents the mean ± SEM for the number of mice indicated on the figure. (**B**) In the zero maze, the time spent in the open portions of the maze is an inverse measure of anxiety-like behavior. The effect of genotype approached statistical significance (*p* = 0.084) so we proceeded with post-hoc t-tests. *Fmr1* KO/*Fxr2* Het mice tended to spend more time in the open portions than WT mice (*p* = 0.065). Each bar represents the mean ± SEM for the number of mice indicated on the figure. ~, 0.05 ≤ *p* ≤ 0.10.

**Figure 4 brainsci-09-00013-f004:**
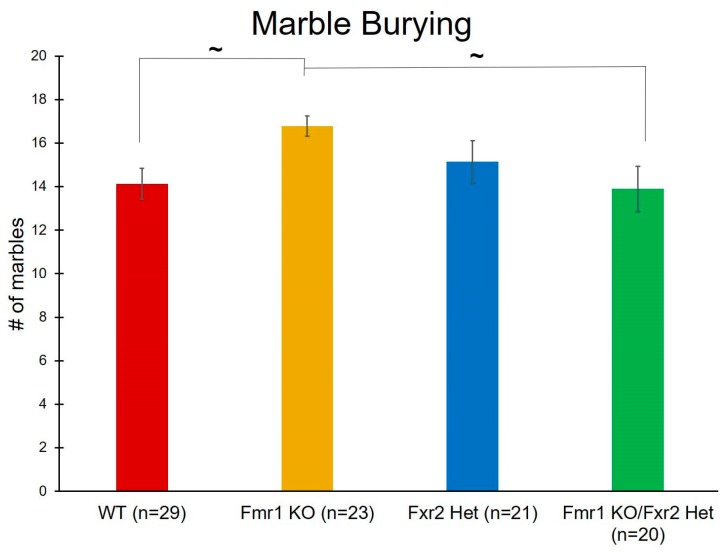
The number of marbles buried is a measure of repetitive behavior. We found a near statistically significant effect of genotype (*p* = 0.052). Post-hoc tests revealed that *Fmr1* knockout (KO) mice tended to bury more marbles than wild-type (WT) (*p* = 0.095) and *Fmr1* KO/*Fxr2* Het (*p* = 0.098) mice. Each bar represents the mean ± SEM for the number of mice indicated on the figure. ~, 0.05 ≤ *p* ≤ 0.10.

**Figure 5 brainsci-09-00013-f005:**
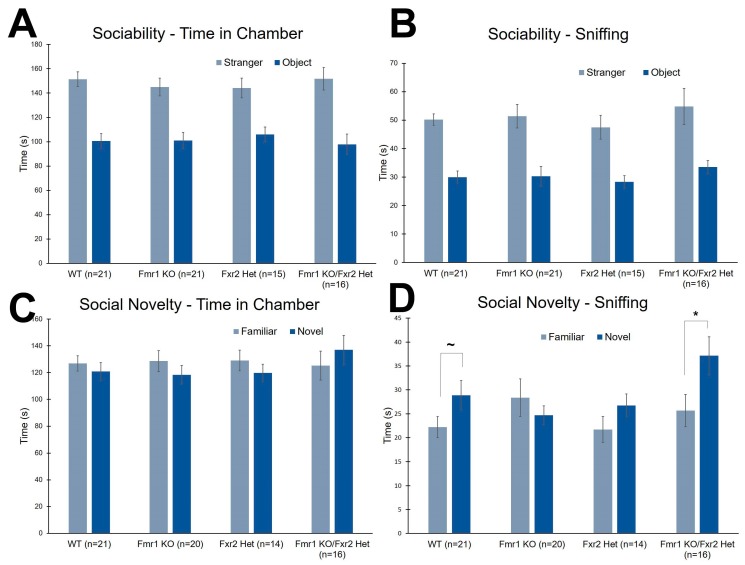
The three-chambered apparatus was used to assess social behavior. (**A**) In the sociability phase, there were no effects of genotype on time in chamber. All mice spent more time in the chamber with the stranger mouse compared to the chamber with the object. (**B**) In the sociability, phase, there were also no effects of genotype on time spent sniffing either the object or the stranger mouse. All mice sniffed the stranger mouse statistically significantly more than the object. (**C**) In the preference for social novelty phase, there were no effects of genotype on time in chamber. None of the four genotypes showed a preference for either the chamber with the novel mouse or the chamber with the familiar mouse. (**D**) In the preference for social novelty phase, there was a statistically significant genotype x chamber interaction for sniffing time (*p* = 0.043). WT and *Fmr1* KO/*Fxr2* Het mice spent more time sniffing the novel mouse compared with the familiar mouse (*p* = 0.062, and *p* = 0.006, respectively), whereas *Fmr1* KO and *Fxr2* Het mice did not. Each bar represents the mean ± SEM for the number of mice indicated on the figure. *, *p* < 0.05. ~, 0.05 ≤ *p* ≤ 0.10.

**Figure 6 brainsci-09-00013-f006:**
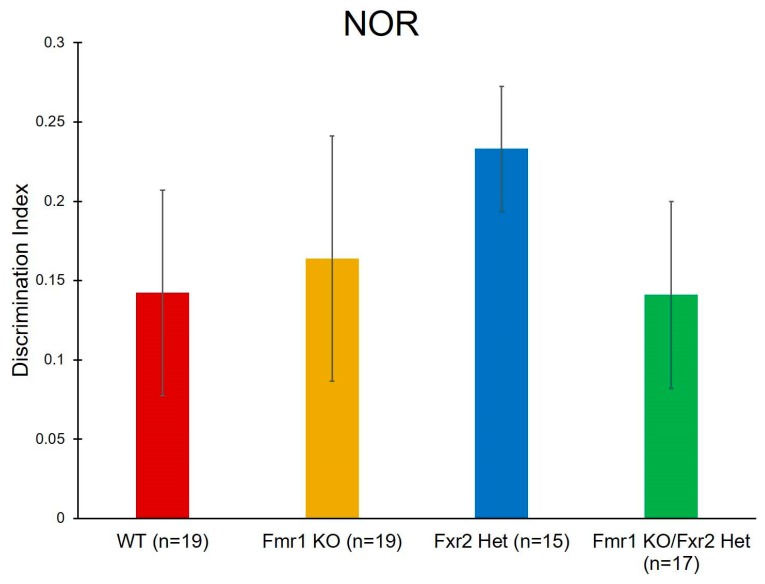
To assay learning and memory, we performed novel object recognition (NOR). We did not find any statistically significant effects. Bars represent the means ± SEMs for the number of mice indicated in parentheses.

**Figure 7 brainsci-09-00013-f007:**
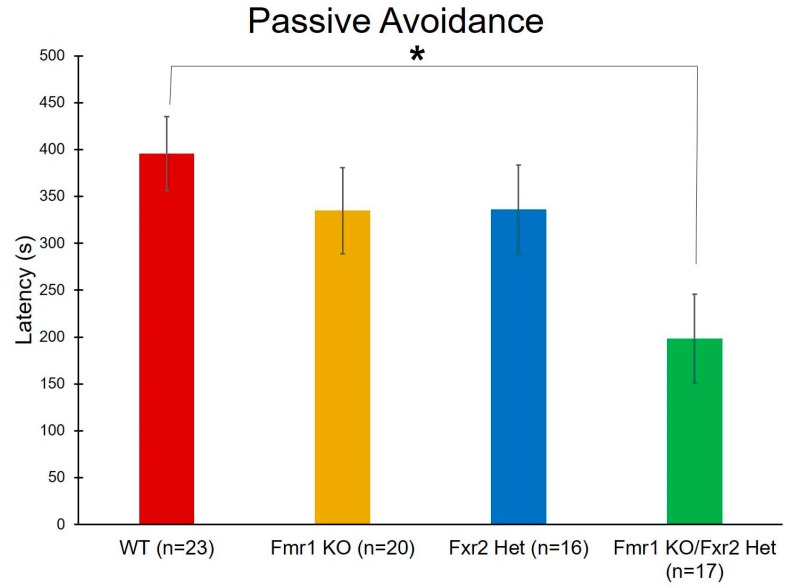
To assay learning and memory, we performed the passive avoidance test. We found a statistically significant effect of genotype on latency to enter the dark chamber (*p* = 0.022). Post-hoc tests revealed that *Fmr1* KO/*Fxr2* Het mice had statistically significantly shorter latencies to enter the dark than WT mice (*p* = 0.014), indicating a deficit in memory. Each bar represents the mean ± SEM for the number of mice indicated on the figure. *, *p* < 0.05.

**Table 1 brainsci-09-00013-t001:** Results of repeated measures ANOVA for behavior testing with corresponding F-values and *p*-values. *, *p* < 0.05. ~, 0.05 ≤ *p* ≤ 0.10.

ANOVA Results				
Behavior	Interaction	Main Effect	F_(*df*,error)_ Value	*p*-Value
**Open Field**				
Distance	Genotype × epoch		F_(*14*,300)_ = 0.796	0.671
		Genotype	F_(*3*,66)_ = 7.477	<0.001 *
		Epoch	F_(*5*,300)_ = 105.254	<0.001 *
Center/Total Distance	Genotype × epoch		F_(*13*,279)_ = 2.390	0.005 *
		Genotype	F_(*3*,66)_ = 13.296	<0.001 *
		Epoch	F_(*4*,279)_ = 2.034	0.074 ~
**Zero Maze**		Genotype	F_(*3*,92)_ = 2.283	0.084 ~
**Marble Burying**		Genotype	F_(*3*,89)_ = 2.675	0.052 ~
**Sociability**				
Time in Chamber	Genotype × chamber		F_(*3*,69)_ = 0.252	0.860
		Genotype	F_(*3*,69)_ = 0.236	0.871
		Chamber	F_(*1*,69)_ = 47.857	<0.001 *
Sniffing Time	Genotype × chamber		F_(*3*,67)_ = 0.026	0.994
		Genotype	F_(*3*,67)_ = 1.186	0.322
		Chamber	F_(*1*,67)_ = 53.519	<0.001 *
**Social Novelty**				
Time in Chamber	Genotype × chamber		F_(*3*,69)_ = 0.427	0.734
		Genotype	F_(*3*,69)_ = 1.981	0.125
		Chamber	F_(*1*,69)_ = 0.206	0.651
Sniffing Time	Genotype × chamber		F_(*3*,67)_ = 2.871	0.043 *
		Genotype	F_(*3*,67)_ = 1.516	0.218
		Chamber	F_(*1*,67)_ = 6.344	0.014 *
**NOR**		Genotype	F_(*3*,70)_ = 0.519	0.671
**Passive Avoidance**		Genotype	F_(*3*,72)_ = 3.421	0.022 *

**Table 2 brainsci-09-00013-t002:** Results of post-hoc *t*-tests, Bonferroni corrected, following statistically significant genotype × epoch interaction in the ratio of center distance to total distance in the open field. *, *p* < 0.05. ~, 0.05 ≤ *p* ≤ 0.10.

*p*-Values for Post Hoc Pairwise Comparisons—Center: Total Distance Ratio Open Field						
Epoch	WT × *Fmr1*	WT × *Fxr2*	WT × *Fmr1/Fxr2*	*Fmr1* × *Fxr2*	*Fmr1* × *Fmr1/Fxr2*	*Fxr2* × *Fmr1/Fxr2*
1	0.973	1.000	0.085 ~	1.000	1.000	1.000
2	0.036 *	1.000	<0.001 *	0.312	0.567	0.006 *
3	0.261	1.000	<0.001 *	1.000	0.031 *	0.003 *
4	0.014 *	1.000	0.001 *	0.710	1.000	0.128
5	0.001 *	1.000	0.003 *	0.006 *	1.000	0.010 *
6	<0.001 *	1.000	<0.001 *	0.005 *	1.000	0.005 *

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
