# Peer review of "Comparative Behavioral Phenotypes of Fmr1 KO, Fxr2 Het, and Fmr1 KO/Fxr2 Het Mice"

_brainsci, 2019, doi:10.3390/brainsci9010013_

Round 1
Reviewer 1 Report
Saré et al. describe work to characterize behaviors in C57/Bl6 mice lacking one copy of Fxr2 with or without the fragile X protein FMRP. The manuscript is well powered, well written and well presented. The data are well and thoroughly presented and support the conclusions drawn. The conclusions are generally in line with prior work that suggests compensation between these two members of the FXR gene family is present in mice. The authors find effects of Fxr2 heterozygosity in Fmr1 KO that are not always consistent (sometimes they exacerbate, sometimes they reduce phenotypes). While they provide no explanation for these differences, these observations are fascinating and can undergird new hypotheses for functional interactions that may differ between cells, circuits, developmental stages, etc to produce changed behaviors. There is some fairly useful supporting literature that might be cited to help flesh out the potential explanations. Work from Justin Fallon on FXR granules that have developmental and cell compartment profiles comes to mind. It is clear that a readout of a complex behavior is complicated by many factors and the path from missing a protein to social behaviors is a lengthy one.
The manuscript appears to have a limitation on the number of references. There are quite a few studies that might be cited in addition. For example, there are clinical reports of individuals with mutations (deletions) that include Fxr1 or Fxr2 along with adjacent genes. The role of the FXRs in these patients’ phenotypes is therefore unclear, but the statement about this in the manuscript is incorrect. Also, Fxr1 is at least equally expressed with the other two FXRs at the mRNA level in brain tissue. It has additional expression in muscle. The manuscript downplays Fxr1’s role in the CNS. The work from Lumaban et al. describing effects of Fmr1/Fxr2 double knockout on metabolism includes Fxr2 het data and might be instructive for understanding the compensation effects.
Author Response
We have added discussion of FXR granules by Justin Fallon.
We have removed the statement about the lack of clinical evidence for FXR2 mutations and added some clinical examples.
We have added that FXR1 is expressed in the brain, but additionally expressed in muscle and cardiac cells.
We have added discussion of the metabolism phenotype of the Fxr2 KO and Hets described by Lumaban.
Reviewer 2 Report
This manuscript reports the effects of heterozygous deletion of the FMRP autosomal paralog FXR2P on behavior in Fmr1 KO mice. The results are presented clearly, and, for the most part, conclusions are supported by the data. Limitations are discussed adequately. I have a few comments and suggestions to improve the manuscript:
1) The authors should show by qRT-PCR or western blotting that heterozygous deletion of FXR2P indeed leads to significant reduction of the protein.
2) The authors discuss a lot of "trending" data. Given the (very rigorously reported) limitations of the study, e.g. the fact that a male and a female experimenter performed behavioral assays and that mouse cohorts/genotypes were not evenly split between the two, discussing these trends may be an overinterpretation of inconclusive data. I suggest to reduce discussion of the "trends".
3) The number of n of mice differ between experiments suggesting that for different behavioral assays, several mice were "outliers". The number of outliers should be reported for each behavior.
4) Related to my last two comments, the authors could test whether their study was sufficiently powered, which would provide insight into the importance of the trending data.
5) The authors should consider showing the novel object data or at least briefly discuss why they think the results were so variable.
6) In the introduction, the authors should also mention the large body of work showing additional functions of FMRP, apart from being a translational regulator.
7) It would be great to provide more justification to why the FXr2p het mouse model was chosen.
8) Methods: When talking about posthoc t-tests in ther statistics section (2.9) the authors should mention that these were Bonferroni corrected, as indicated in one of the figure legends.
9) A brief discussion of where FMRP and FXR2P are expressed in relation to each other in the brain would be helpful and could be put into context with their "paradoxical" behavioral data.
Author Response
1) The authors should show by qRT-PCR or western blotting that heterozygous deletion of FXR2P indeed leads to significant reduction of the protein.
We have now added western blots showing that FXR2P is reduced by 60% in heterozygous animals.
2) The authors discuss a lot of "trending" data. Given the (very rigorously reported) limitations of the study, e.g. the fact that a male and a female experimenter performed behavioral assays and that mouse cohorts/genotypes were not evenly split between the two, discussing these trends may be an overinterpretation of inconclusive data. I suggest to reduce discussion of the "trends".
Several instances referring to statistically trending data have been removed from the manuscript.
3) The number of n of mice differ between experiments suggesting that for different behavioral assays, several mice were "outliers". The number of outliers should be reported for each behavior.
We have added exactly how many animals of each genotype in each test were not included and why.
4) Related to my last two comments, the authors could test whether their study was sufficiently powered, which would provide insight into the importance of the trending data.
For some behavior tests like open field testing, in which the phenotype is robust and variability is low power was adequate. Based on our previous studies in which we used the passive avoidance test (Qin et al. Brain, Behavioral Research, 2015; Fig. 4C) power was adequate. We should have been able to see a difference (P<0.05, 95% power) between WT and Fmr1 KO mice with 10 mice per group. We found much greater variability in the present study as discussed in the paper. Based on the studies of others (Seese et al., PNAS, 2014; Fig. 1I) we should have been able to detect a difference in performance between WT and Fmr1 KO mice with 13 per group (P<0.05, 95% power). In our hands variability was much greater than previously reported and no statistically significant differences were detected. For the NOR, we don’t know why variability is so great in our hands.
5) The authors should consider showing the novel object data or at least briefly discuss why they think the results were so variable.
We have added the NOR data.
6) In the introduction, the authors should also mention the large body of work showing additional functions of FMRP, apart from being a translational regulator.
We discuss additional functions ascribed to FMRP.
7) It would be great to provide more justification to why the FXr2p het mouse model was chosen.
We have expanded upon the justification to study the het model.
8) Methods: When talking about posthoc t-tests in ther statistics section (2.9) the authors should mention that these were Bonferroni corrected, as indicated in one of the figure legends.
We have added this detail to the methods section.
9) A brief discussion of where FMRP and FXR2P are expressed in relation to each other in the brain would be helpful and could be put into context with their "paradoxical" behavioral data.
Studies seem to place the expression of FMRP and FXR2P in the same regions of the brain. We have added that these genes are overlapping in their expression pattern. However, we have added some discussion about possible differing roles of these proteins on a molecular level.
Round 2
Reviewer 2 Report
The authors responded adequately to most of my comments. There are two minor points that should be addressed:
1) FMRP was shown to directly bind to ion channels and be involved in DNA damage response. These functions should be mentioned in addition to the fact that it is a nuclear shuttle protein.
2) The new western blot in Figure 1 (1A) should be labeled to indicate the genotypes of the samples and which protein was detected. With the current presentation, the reader has to read the figure legend to draw conclusions. In addition, the loading control should be shown (I assume the signal was normalized to a loading control, as the y axis is labeled with "relative expression".
Author Response
The authors responded adequately to most of my comments. There are two minor points that should be addressed:
1) FMRP was shown to directly bind to ion channels and be involved in DNA damage response. These functions should be mentioned in addition to the fact that it is a nuclear shuttle protein.
We have expanded on the functions of FMRP to include these as well.
2) The new western blot in Figure 1 (1A) should be labeled to indicate the genotypes of the samples and which protein was detected. With the current presentation, the reader has to read the figure legend to draw conclusions. In addition, the loading control should be shown (I assume the signal was normalized to a loading control, as the y axis is labeled with "relative expression".
We have labeled the genotypes on the blot. Also, we added more detail about the Western blotting to the methods section. We did not normalize to a housekeeping protein, but we normalized to total protein loaded by employing the Bio-Rad stain-free gel technology. This method is particularly important in a disorder like Fragile X which is thought to affect protein translation.